# Electrocatalytic Oxidation of Glucose on Boron and Nitrogen Codoped Graphene Quantum Dot Electrodes in Alkali Media

**Siyong Gu** [1] , **Chien-Te Hsieh** [2,3,*] , **Chih-Peng Kao** [4], **Chun-Chieh Fu** [4], **Yasser Ashraf Gandomi** [5] , **Ruey-Shin Juang** [4,6,*] **and Kenneth David Kihm** [3,*]

1   Fujian Provincial Key Laboratory of Functional Materials and Applications,
    School of Materials Science and Engineering, Xiamen University of Technology, Xiamen 361024, China;
    gu-siyong@163.com
2   Department of Chemical Engineering and Materials Science, Yuan Ze University, Taoyuan 32003, Taiwan
3   Department of Mechanical, Aerospace, and Biomedical Engineering, University of Tennessee,
    Knoxville, TN 37996, USA
4   Department of Chemical and Materials Engineering, Chang Gung University, Guishan, Taoyuan 33302,
    Taiwan; kevinkao1995911@gmail.com (C.-P.K.); charles07172003@gmail.com (C.-C.F.)
5   Department of Chemical Engineering, Massachusetts Institute of Technology, Cambridge, MA 02142, USA;
    ygandomi@mit.edu
6   Division of Nephrology, Department of Internal Medicine, Chang Gung Memorial Hospital,
    Linkou 33305, Taiwan
*   Correspondence: cthsieh@saturn.yzu.edu.tw (C.-T.H.); rsjuang@mail.cgu.edu.tw (R.-S.J.);
    kkihm@utk.edu (K.D.K.)

**Abstract:** A novel solvothermal technique has been developed in the presence of C/N/B precursor for synthesizing B-N-coped graphene quantum dots (GQDs) as non-metal electrocatalysts towards the catalytic glucose oxidation reaction (GOR). Both N-doped GQD and B-N-codoped GQD particles (~4.0 nm) possess a similar oxidation and amidation level. The B-N-codoped GQD contains a B/C ratio of 3.16 at.%, where the B dopants were formed through different bonding types (i.e., N-B, C-B, $BC_2O$, and $BCO_2$) inserted into or decorated on the GQDs. The cyclic voltammetry measurement revealed that the catalytic activity of B-N-codoped GQD catalyst is significantly higher compared to the N-doped GQDs (~20% increase). It was also shown that the GOR activity was substantially enhanced due to the synergistic effect of B and N dopants within the GQD catalysts. Based on the analysis of Tafel plots, the B-N-codoped-GQD catalyst electrode displays an ultra-high exchange current density along with a reduced Tafel slope. The application of B-N-codoped GQD electrodes significantly enhances the catalytic activity and results in facile reaction kinetics towards the glucose oxidation reaction. Accordingly, the novel design of GQD catalyst demonstrated in this work sets the stage for designing inexpensive GQD-based catalysts as an alternative for precious metal catalysts commonly used in bio-sensors, fuel cells, and other electrochemical devices.

**Keywords:** non-metal catalysts; graphene quantum dots; glucose oxidation; boron doping; catalytic activity; nitrogen doping

## 1. Introduction

Nanostructured materials are an emerging class of nanomaterials that have gained significant attention since their physiochemical characteristics can be engineered and even enhanced compared to their corresponding bulk material properties [1]. In particular, nanostructured materials enable higher surface areas and enhanced surface energy [2,3]. The carbon-based nanostructured materials have been widely used for energy storage and sensing devices due to superior conductivity, biocompatibility, and higher surface areas [4–7].

Among various classes of nanostructured materials, the graphene quantum dots (GQDs) have received significant attention recently [8]. The capability of adjusting the

bandgap within a relatively wide range through adjusting the size and structure of GQDs [9] along with substantial quantum confinement effects enable engineering the GQDs for improving the physicochemical properties [10,11]. Major advantages of GQDs include high solubility at room temperature, robust photoluminescence, and negligible cytotoxicity [10,11]. Prior successful implementations of GQDs have been focused on sensors with superior selectivity [12,13], optoelectronic devices [14], bio-imaging [15], ionic detection [16], and high-performance redox flow batteries [17,18].

For many electrochemical devices, a catalyst is needed for deriving the oxidation and reduction reactions on the anode and cathode electrodes, respectively. In particular, electrooxidation of glucose ($C_6H_{12}O_6$, molecular weight: 180 g mol$^{-1}$) is of significant importance for many electrochemical energy conversion devices (e.g., glucose fuel cells) [19,20]. It is well established that the catalyst choice directly influences the reaction pathway during the glucose oxidation reaction (GOR). It is also shown that the oxidation of glucose molecule to carbon dioxide is capable of attaining a high theoretical energy (ca. 2870 kJ mol$^{-1}$) via liberating 24 electrons [21–24].

The electrochemical GOR can be classified into two major categories including enzymatic and non-enzymatic reactions [1]. Pioneering studies have shown that the systems/devices employing non-enzymatic GOR (usually with metal oxides as the catalyst and nanostructured porous metals or carbon materials as the electrode) have a longer lifetime and are more stable compared to the enzymatic sensors [25,26]. Many attempts have been devoted for exploring the non-enzymatic GOR using precious metals (e.g., Pt [19], Au [24,27], and Pd [28]) and metal oxides (e.g., $Co_3O_4$ [29] and $NiFeO_x$ [30]) as the nano-catalysts. Engineering the electrodes' microstructure, tuning the roughness factor, and increasing the active surface area have also been explored for boasting the catalytic activity of the GOR [25–30].

Prior works on improving the catalytic behavior of GOR have predominantly focused on deploying precious metals as the catalyst. Despite being very effective, this class of catalysts is too expensive for widespread adoption. Therefore, there is a significant need for developing inexpensive yet high-performance catalysts for the glucose oxidation reaction. In this work, we report, for the first time, the fabrication and characterization of B-N-codoped GQD electrodes as a non-enzymatic catalyst for the GOR. Doping N within the graphene imparts *n*-type conductivity [31] while B-doping leads to *p*-type conductivity in the graphene nanostructure due to the low electronegativity of the boron atom (even less than carbon; 2.04 for B vs. 2.55 for C [32]). To synthesize the B-N-codoped GQDs, a solvothermal route was developed for effectively tuning the electronic structure of the catalytic electrodes. For comparison, we have also synthesized N-doped GQDs employing the solvothermal technique developed in-house. The catalytic GOR on as-prepared GQD catalysts was analyzed using cyclic voltammetry (CV) at a high-pH environment. The stability as well as the catalytic activity of the GQD electrodes was also analyzed for the non-enzymatic GOR.

## 2. Results and Discussion

### 2.1. Physiochemical Properties of B-N-Codoped GQDs

The TEM micrograph of N-doped GQD sample, as illustrated in Figure 1a,b, reveals a homogeneous dispersion of nanoparticles. The N-doped GQD sample displays a quasi-spherical shape with an average diameter of ca. 4.0 nm. We also observe well-ordered lattice fringes with an average spacing distance of ca. 0.21 nm corresponding to the (100) facet of graphite. Such an ordered lattice morphology confirms the formation of graphite-like structure for the GQD samples [33]. The TEM micrographs of the B-N-codoped GQDs are provided in Figure 1c,d. According to Figure 1c,d, the B-N-codoped GQDs exhibit a spherical-like shape after the introduction of B dopant. The lattice fringes of B-N-codoped GQD samples are localized within several small domains due to the presence of grain boundaries formed upon doping B and N atoms. Indeed, the boron doping alters the lattice

arrangement due to the substitutional or interstitial replacement on the basal planes or at the edge sites of GQDs.

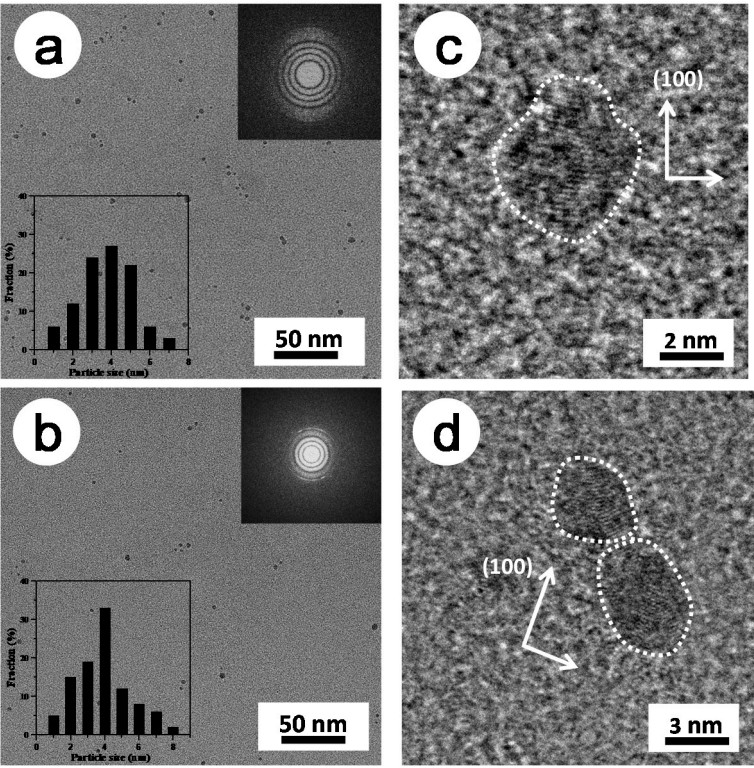

**Figure 1.** TEM micrographs of (**a**) N-doped graphene quantum dots (GQD) and (**b**) B-N-codoped GQD samples. The corresponding histograms have also been shown within the insets. The lattice fringes of (**c**) N-doped GQD and (**d**) B-N-codoped GQD samples.

The corresponding selected-area diffraction patterns of the GQD nanomaterials are shown at the inset of Figure 1a,c. Based on Figure 1a,c, several ring-like structures can be identified exhibiting typical features of a hexagonal graphite lattice [34]. Nonetheless, the incomplete rings of both GQD samples reflects the disordered structure of carbon lattices imposed through the structural defects caused by the B and N doping [35]. The histograms for the particle size distributions of both samples are illustrated in the inset of Figure 1a,c; revealing a quasi-Gaussian distribution centered at 3–5 nm. The particle size distributions for both GQD samples were analyzed by counting more than 100 nanoparticles. The particle size distributions were basically fitted to the log-normal distribution. The average particle sizes of N-doped GQD and B-N-codoped GQD samples were estimated to be 4.0 and 3.9 nm, respectively. This observation implies that the influence of B doping on the particle shape as well as the particle size is miniscule.

Typical XRD patterns of both GQD samples are given in Figure 2, demonstrating a diffraction peak at $2\theta = 23\text{-}25°$, assigned to the (002) plane of graphite [36]. For the graphitic crystalline structures, it is widely accepted that the Bragg's equation can be used for assessing the interlayer distance ($d_{(002)}$) [37]. Usually, the (002) peak of artificial graphite takes place at ca. $26.4°$ ($d_{(002)} = 0.337$ nm), which is slightly lower than that of both GQD samples ($d_{(002)} = 0.356\text{-}0.386$ nm). The larger interlayer spacing distance is attributed to the presence of nitrogen and boron dopants inserted into the graphitic lattices. For the B-N-codoped GQD sample, we observe representative peaks of boron carbide, based on JCPDS card no. 35-0798 [38]. The other three diffraction peaks can be attributed to the cubic crystalline $B_2O_3$, according to JCPDS card no. 06-0297 [39]. This finding reveals that B-N-codoped GQDs contain B and N dopants (in the graphitic lattices) along with boron oxides attached to the edge of the lattices or embedded within the lattices.

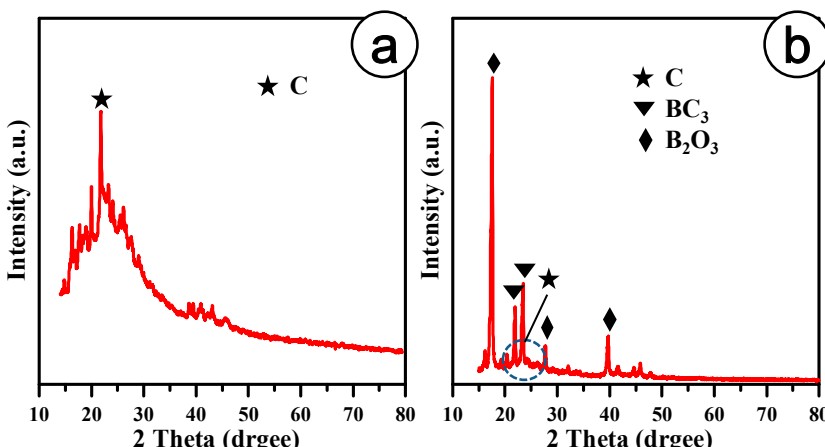

**Figure 2.** Typical XRD patterns of (**a**) N-doped GQD and (**b**) B-N-codoped GQD samples.

The XPS measurements were adopted to analyze the variations in the dopant concentration as well as distribution of surface functionalities induced by the B and N doping. The elemental analysis of C, N, O, and B can be explored through focusing on different binding energies: C 1s (ca. 282-288 eV), N 1s (ca. 399-408 eV), O 1s (ca. 531-536 eV), and B 1s (ca. 190-194 eV) [40–42]. Indeed, the N/C atomic ratio for the N-doped GQD and B-N-codoped GQD samples was identical (ca. 13.3 at.%), while both GQD samples contained a similar O/C atomic ratio, i.e., 22.2 at.% (N-doped GQDs) and 21.6 at.% (B-N-codoped GQDs). This result reflects that both GQD samples included comparable amidation and oxidation levels. It is important to note that the B/C ratio within the samples demonstrated an increasing trend up to 3.16 at.%, indicating that the B dopants (originating from boric acid during the solvothermal process) were successfully inserted within the GQD lattices; an observation also confirmed by the XRD analysis, as discussed previously.

Detailed-scan XPS spectra of N 1s, O 1s, and N 1s peaks of N-doped GQD samples, decomposed using a multiple Gaussian function, are depicted in Figure 3a–c. According to Figure 3a, the C 1s spectrum can be deconvoluted into three main peaks including C=C/C-C (ca. 284.5 eV), C-N/C-OH (ca. 286.0 eV), C=O (ca. 286.8 eV) [43,44]. This observation reveals that the oxygen and nitrogen functional groups were successfully attached to the edges or basal planes of graphite-like lattices. In addition, the N 1s spectrum was comprised of two components assignable to pyridinic/pyrrolic nitrogen (ca. 399.6 eV) and $NO_2$ (ca. 406.5 eV) [45], as illustrated in Figure 3b. Herein, the pyridinic/pyrrolic N has a higher tendency to attach to the edge sites ($C_5$ or $C_6$) rather than at basal plane sites of GQD lattices [46–48]. The O 1s spectrum consists of two major components: C=O (ca. 531.8 eV) and C–O/C–$NO_2$ (ca. 533.5 eV), as shown in Figure 3c [45]. Therefore, the N-doped GQDs contain oxygen functional groups, lattice N atoms, and $NO_2$ groups attached to the edge sites.

Figure 4a–d presents high-resolution C 1s, N 1s, B 1s, and O 1s peaks of the B-N-codoped GQD samples. Based on Figure 4a, one additional group (i.e., C-B at ca. 283.9 eV [49,50]) within the C 1s peak can be identified that was not present in the N-doped GQDs spectra. This additional peak suggests the existence of B dopants in the GQD lattice; an observation also confirmed through the XRD analysis. As shown in Figure 4b, the N 1s spectrum consists of three components: N-B bonding (ca. 398.3 eV), graphitic N (ca. 401.1 eV), and $NO_2$ (ca. 406.5 eV) [45,51]. The B 1s spectrum of B-GQD-13 (see Figure 4c) exhibits three major peaks around ca. 189.8, 190.6, and 191.9 eV, corresponding to C-B, $BC_2O$, and $BCO_2$ bonds, respectively [52]. Therefore, it can be concluded that the B atoms were successfully doped within the skeleton of GQDs. It is important to note that the intense peak at ca. 190.6 eV can be assigned to the structure of B atoms bonding to C and O atoms ($BC_2O$), resulting in the discharge of boron atoms from the nanostructure in the form of epoxy. The signal at ca. 191.9 eV reveals that B atoms are surrounded by C and O atoms ($BCO_2$), revealing the appearance of boronic acid group in B-N-codoped

GQDs [53]. As depicted in Figure 4d, the O 1s spectrum confirms the presence of C=O and C–O/B-O/C–NO$_2$ bonds at ca. 531.8 and 533.5 eV, respectively [45,54,55]. As a result, it can be inferred that the B dopants are predominantly formed through four different bonding types (i.e., N-B, C-B, BC$_2$O, and BCO$_2$) inserted into or decorated on the GQDs structure.

### 2.2. GOR on B-N-Codoped GQD Electrodes

Figure 5 includes a schematic diagram demonstrating the synthesis procedure (solvothermal) of the B-N-codoped GQDs. As shown in Figure 5, the solvothermal route includes three major steps: (i) Activation of pyrene precursor in the presence of nitric acid, (ii) solvothermal synthesis from the TNP + boric acid mixture, and (iii) electrode assembly for GOR.

To explore the efficacy of as-prepared GQD catalytic electrodes, the CV was performed in 1 M glucose + 1 M NaOH at the scan rate of 50 mV s$^{-1}$, as shown in Figure 6a,b. The CV spectra of GQD electrodes were collected within the potential range of −0.6–0.4 V vs. Ag/AgCl. The CV curves associated with the glucose oxidation reaction using as-prepared catalytic electrodes (as the working electrode) include two oxidation peaks (see Figure 6) similar to the CV graphs of GOR reported in the literature with precious metal catalysts (e.g., Pd-Ru [23]). According to Figure 6, both GQD catalyst electrodes exhibit a relatively similar oxidation onset potential (−0.50 V vs. Ag/AgCl for the B-N-codoped GQD and −0.48 V for the N-doped GQD). Indeed, the chemisorption of hydroxyl ions onto the GQD catalysts in the initial stage of oxidation results in the formation of GQD–OH$_x$ sites [3]. On the surface of B-N-doped GQDs, the N sites are usually occupied by the oxygen compounds, whereas the B sites promote the stripping of originally oxygenated species (e.g., −OH), due to the higher electro-negativity of N compared to B atoms (~3.04 (N) vs. ~2.04 (B)). This mechanism has also been reported for the improved catalytic activity of glucose oxidation reaction using binary Pt-Ni [56] and Pt-Sn [57] catalysts. Accordingly, the initial reaction can be written as:

$$\text{GQD} + x\,\text{OH}^- \rightarrow \text{GQD-OH}_x + x\,\text{e}^- \tag{R1}$$

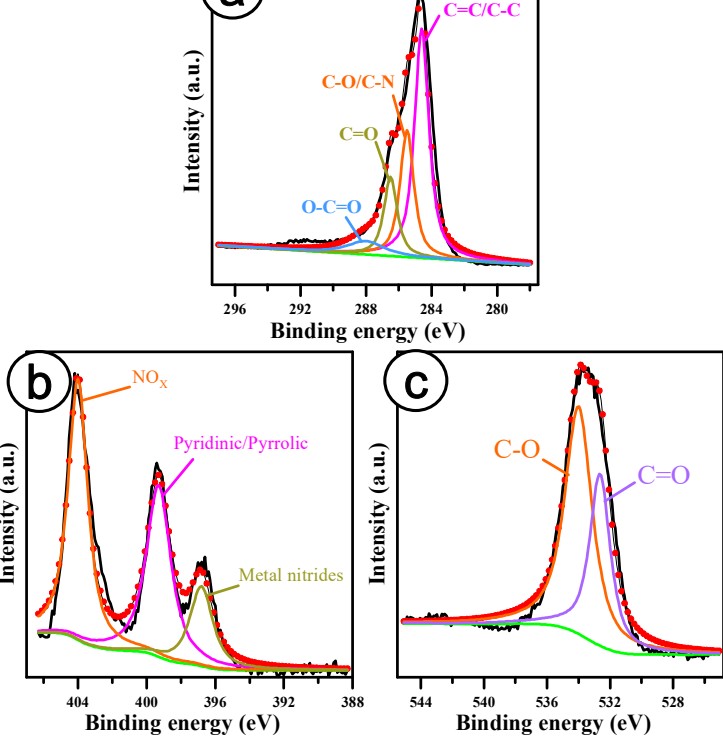

**Figure 3.** XPS spectra of (**a**) C 1s, (**b**) N 1s, and (**c**) O 1s peaks of N-doped GQD sample.

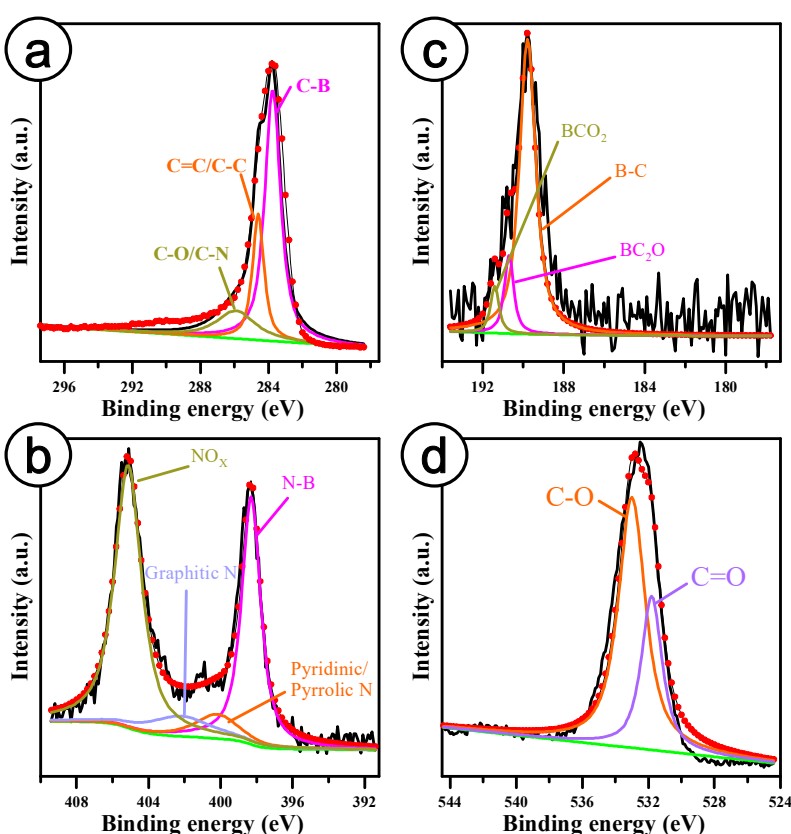

**Figure 4.** XPS spectra of (**a**) C 1s, (**b**) N 1s, (**c**) B 1s, and (**d**) O 1s peaks of B-N-codoped GQD = coated electrodes.

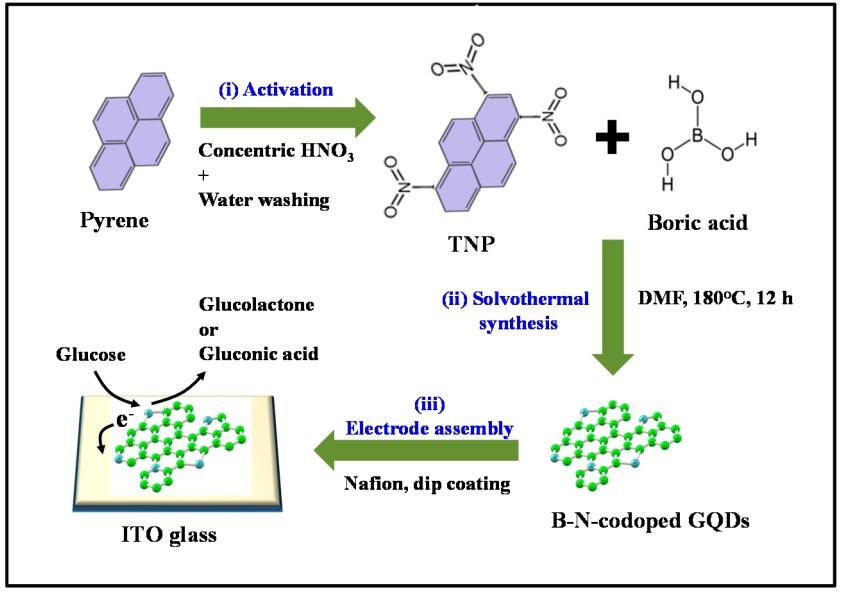

**Figure 5.** Schematic diagram of synthesizing B-N-codoped GQDs through the solvothermal synthesis method for glucose oxidation.

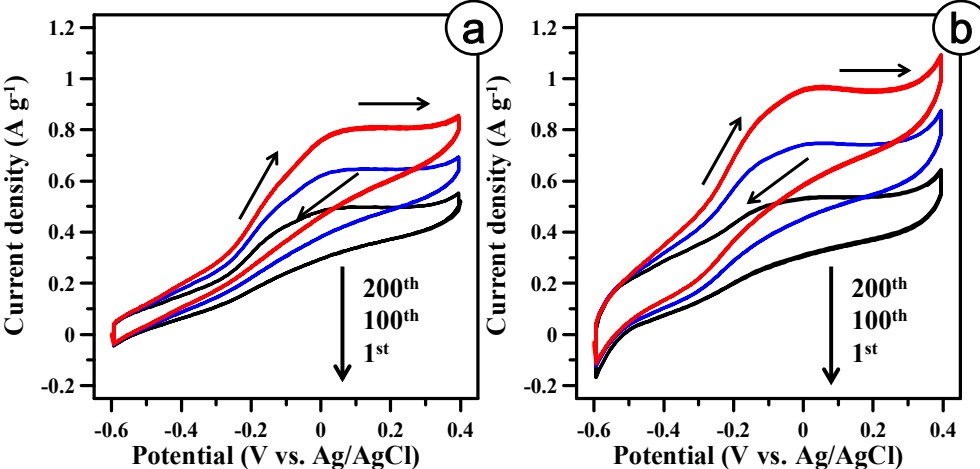

**Figure 6.** Typical cyclic voltammetry (CV) profiles of (**a**) N-doped GQD and (**b**) B-N-codoped GQD electrodes in 1 M glucose + 1 M NaOH electrolyte at 50 mV s$^{-1}$.

The onset potential can usually be correlated to the electrochemical activity of an electro-catalyst; lower than the onset potential, better than the catalytic activity [58]. Based on the CV analysis, we observe that both as-prepared catalysts display a similar onset potential towards the glucose oxidation reaction, which is very close to the onset potential reported for the GOR using precious Pd-Rh nanocatalysts [23]. The improved catalytic activity enabled with the as-prepared GQD catalysts is majorly due to the highly crystalline structure of the nanocatalysts (i.e., fast charge-transfer rate), small particle size (i.e., average particle size: ~4 nm), and high amidation level (i.e., N/C ratio: ~13.3 at.%).

As illustrated in the CV profiles (see Figure 6), the initial oxidation peak attributed to the direct GOR is observed within the potential range of $-0.1$ to 0.1 V vs. Ag/AgCl (in the positive scan). It is well established that the electro-oxidation of glucose is usually difficult and has intrinsically sluggish kinetics [1,59]. The first step of GOR is the electro-chemical adsorption of the glucose molecules on the surface of the GQD electrode during dehydrogenation (i.e., the formation of dehydrogenated glucose (GQD-glucose)), as shown in the following:

$$GQD\text{-}OH_x + glucose \leftrightarrow GQD\text{-}glucose + H_2O + e^- \tag{R2}$$

Second, the dehydrogenated molecule is transformed to gluconate through direct oxidation (i.e., R3), which involves the generation of a hydroxide ion and elimination of a proton (H$^+$). The oxidization of the dehydrogenated glucose to gluconolactone (i.e., R4) is also feasible as an alternative pathway for the reaction. The gluconolactone is then transformed to gluconate upon reacting with a hydroxide ion [27].

$$GQD\text{-}glucose + OH^- \leftrightarrow GQD\text{-}gluconate + H^+ + e^- \tag{R3}$$

$$GQD\text{-}glucose \leftrightarrow GQD\text{-}gluconolactone \ (or \ gluconate) + H^+ + e^- \tag{R4}$$

Shifting the potential to more positive potentials, the GQD–OH$_x$ sites tend to catalyze the dehydrogenated glucose (i.e., GQD-glucose) [60], thus, inducing the first oxidation peak in the positive scan. In the negative potential scan, the reduction of oxidized GQD sites occurs within the potential range of $-0.2$–0 V (vs. Ag/AgCl). Subsequently, the occupied sites are electrochemically stripped as the active sites for the glucose oxidation reaction, as featured in the second oxidation peak.

The first oxidation peaks associated with the reactions formulated through R2–R4 can be used for evaluating the catalytic activity of the as-prepared GQD electrodes [60]. According to Figure 6a,b, the specific current evaluated at the 1st cycle demonstrates the following order: B-N-codoped GQD (0.98 A g$^{-1}$) > N-doped GQD (0.81 A g$^{-1}$). This analysis reveals

that the catalytic activity of B-N-codoped GQD catalyst is 1.21 times higher than that of N-doped GQD electrodes. After 200 cycles, the specific current still maintains: B-N-codoped GQD (0.55 A g$^{-1}$) > N-doped GQD (0.49 A g$^{-1}$). The increased catalytic activity is majorly due to the synergistic effect of B and N dopants in the GQD catalysts [61]. Indeed, the N functionalities facilitate the formation of GQD–OH$_x$ sites, enabling the progress of GOR steps (R2-R4) in the forward direction; whereas the B dopants refresh the occupied and poisoned sites for the next GOR cycle. Using the density functional theory, it became clear that BC$_3$ plays a crucial role in determining the electronic transition, while the influence of BC$_2$O and BCO$_2$ (i.e., oxidized B bonding configurations) on the electronic properties of GQDs mostly depends on their hybridization form of carbon [62]. Therefore, during the catalytic GOR cycle, the glucose molecule interacts with the B center where the electronic properties of the B sites affect the electrocatalytic activity.

Considering the CV curves shown in Figure 6, the 1st peak (i.e., J$_1$, in the positive scan) can be assigned to the catalytic activity during the GOR in the direct direction, while the appearance of the 2nd peak (i.e., J$_2$, in the negative scan) represents the reduction of oxides; an important index in evaluating the availability of pre-occupied sites for the subsequent GOR step upon recovery. Indeed, the J$_1$/J$_2$ ratio can be assessed as a function of the cycle number. The J$_1$/J$_2$ ratio of N-doped GQD catalyst ranges from 1.72 to 1.81, which is higher than that of the B-N-codoped GQD catalyst (1.63-1.71) for the entire 200 cycles. This finding reveals that the presence of B dopants efficiently regenerates the poisoned sites for the subsequent catalytic cycle (i.e., enhanced reversibility), resulting in an improved catalytic activity towards the glucose oxidation reaction.

To confirm the long-term catalytic activity and durability towards GOR, a chronoamperometry test at constant potential of 0 and 0.2 V vs. Ag/AgCl was conducted for a period of 0.5 h. Figure 7 shows the chronoamperometry curves for both catalyst electrodes in an aqueous electrolyte of 1 M glucose + 1 M NaOH. In the initial stage, all potentiostatic currents are found to decrease rapidly, corresponding to the formation of intermediate species such as GQD–OH$_x$, GQD-glucose, GQD-gluconate, and GQD-gluconolactone during the GOR process, resembling the methanol oxidation reaction on metallic catalysts [63,64]. After 0.5 h, the current decay becomes gradual and then remains stable. The steady-state polarization curves reveal that the stable current of B-N-codoped GQD electrode is higher than in the N-doped GQD one. This result indicates that the B-N-codoped GQD catalyst has the highest electrocatalytic activity towards the electrooxidation of glucose and stability, which is identical to the results of CV measurements.

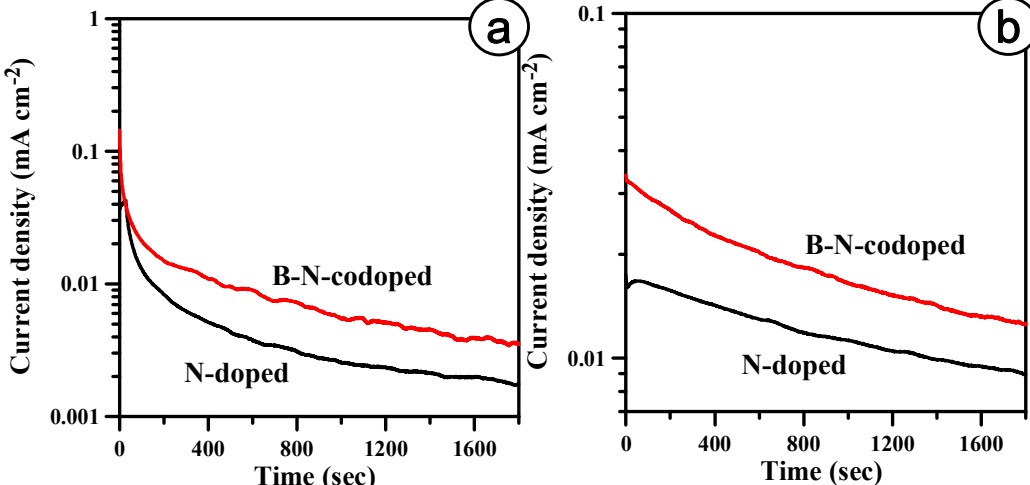

**Figure 7.** Current–time curves for the glucose oxidation reaction (GOR) at (**a**) 0 and (**b**) 0.2 V vs. Ag/AgCl for both catalyst electrodes.

### 2.3. Catalytic Kinetics of B-N-Codoped GQD Electrodes

To characterize the kinetics of the glucose oxidation reaction on different catalytic electrodes, typical Tafel plots were analyzed using the linear sweep voltammetry (LSV) conducted at 50 mV s$^{-1}$. The Tafel plots can be divided into two stages: (i) The positive scan from −0.6 to 0.4 V and (ii) the negative scan from 0.4 to −0.6 V, as shown in Figure 8a,b, respectively. Using the Butler-Volmer kinetic model, the Tafel plots can be formulated as shown in the following [21–23]:

$$\log i = \log i_0 + \left( \frac{\alpha n F}{2.303 RT} \right) \eta \tag{R5}$$

where, $\alpha$ resembles the transfer coefficient; $\eta$, $n$, $R$, and $T$ are the overpotential, Faraday's constant, universal gas constant, and the operating temperature, respectively. Analyzing the polarization curves at low overpotentials (i.e., LSV), the $y$-axis intercept and the slope of the curves can be used for determining the exchange current ($i_0$) and the Tafel slope as illustrated in Figure 8 [30].

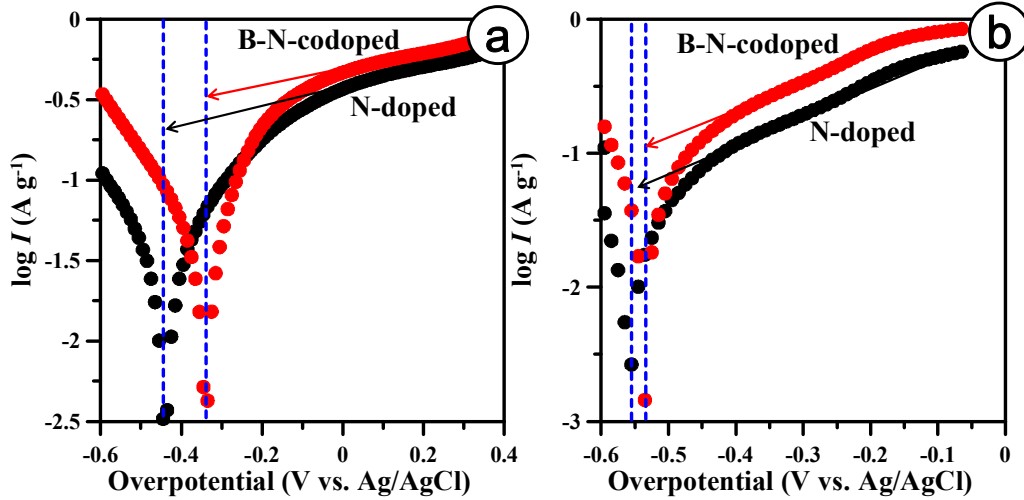

**Figure 8.** Typical Tafel plots of both N-doped GQD and B-N-codoped GQD electrodes in 1 M glucose + 1 M NaOH electrolyte at 50 mV s$^{-1}$: (**a**) The positive scan and (**b**) negative scan. The arrow indicates their corresponding exchange current.

The Tafel slope can be correlated to the electron transfer rate where a smaller Tafel slope implies a rapid electron transfer rate and more favorable catalytic reaction kinetics [65]. The lower Tafel slope of as-prepared GOR confirms a fast electron transfer (i.e., facile kinetics) during the glucose oxidation reaction (stage (i): In the positive scan) and the cleavage of O−H bonds in the H$_2$O molecule to produce O$_2$ (stage (ii): In the negative scan) [30]. According to Figure 7, the Tafel slopes at stage (i) for N-doped GQD and B-N-codoped GQD catalysts were 680 and 672 mV dec$^{-1}$, respectively. A lower Tafel slope is usually the result of a lower adsorption potential between the glucose molecules and the surface of the corresponding electrode [66] and enhanced wettability of the catalysts by the adjacent electrolyte [67]. It is generally recognized that oxygen functionalities are capable of providing a strong affinity to the adsorbed glucose molecules, facilitating the progress of (R2). In the GQD catalysts, the substitution of C atoms with the graphitic N atoms maintains the lattice structure while promoting the sp$^2$ hybridization. Subsequently, such a substitution within the lattice, substantially increases the conductivity of the GQD electrodes [68,69], and enhances the charge transfer rate [70]. In addition, since the GQD catalysts possess high oxidation and amidation levels, the robust design of GQD catalytic electrodes enables a high surface coverage of adsorbed glucose molecules with facile charge transfer kinetics, favoring the glucose oxidation reaction in the forward direction (R2-R4).

According to Figure 8, the Tafel slopes at stage (ii) were assessed to be 2197 and 2033 mV dec$^{-1}$ for the N-doped GQD and B-N-codoped GQD catalysts, respectively. The lower Tafel slope of B-N-codoped GQD catalyst can be attributed to the fact that the B and N dopants enable the desorption of oxygenated intermediate species (e.g., $OH_{(ads)}$) as a result of doping the heteroatom within the lattice. Indeed, the localized distribution of the molecular orbitals is vastly affected by the pyridinic N doping at the edges through the introduction of unpaired electrons and subsequently weakening the $O-O$ bonding [71–73]. Moreover, since the B atoms incorporated within the carbon atoms possess a relative positive charge due to the lower electro-negativity compared to the C atoms [70], stripping the oxygenated species off the B atoms can easily occur owing to the difference in the electro-negativity between the B and O atoms; thus, favoring the subsequent GOR catalytic cycle.

Based on the analysis of the Tafel plots, the magnitude of $i_0$ values among different samples demonstrates the following order: B-N-codoped GQD (0.27 A g$^{-1}$) > N-doped GQD (0.18 A g$^{-1}$) at stage (i) and B-N-codoped GQD (0.11 A g$^{-1}$) > N-doped GQD (0.05 A g$^{-1}$) at stage (ii). The higher $i_0$ values for both stages indicate a faster reaction rate of the catalytic electrodes [73], i.e., B-N-codoped GQD electrode towards the catalytic GOR cycles. This significant improvement mainly originates from the synergistic effect between the N-substitution and B dopant, causing the surface heterogeneity on the GQD lattices and subsequently resulting in the remarkable catalytic activity. The B-N-codoped GQD catalysts synthesized and developed in this work exhibit a very high exchange current while demonstrating a very low Tafel slope, enabling the superior catalytic activity and excellent reaction kinetics towards the glucose oxidation reaction.

## 3. Materials and Methods

### 3.1. Solvothermal Synthesis of B-N-Codoped GQDs

To synthesize the B-N-codoped GQDs, 1,3,6-trinitropyrene (TNP, home-made) precursor was prepared through the nitration of pyrene (Sigma-Aldrich, St. Louis, MO, USA, purity: 98%) in the presence of concentric nitric acid (J.T. Baker, Radnor, PA, USA, purity: 70%) [45,74]. During the nitration process, pyrene (2 g) was mixed in 240 mL concentric HNO$_3$ (16 N) at 80 °C for 18 h using a magnetic bar with the rotational speed of 150 rpm. The as-prepared solution was naturally cooled down to ambient temperature; subsequently, the solution was diluted with distilled water (2 L) and then continuously stirred at ambient temperature for 24 h, enabling the attachment of NO$_x$ groups to the TNP molecules. The as-prepared solution was filtered by using a vacuum filtration apparatus (MF-Millipore™ Membrane Filter, cellulose ester microporous filter, Darmstadt, Germany, pore size: 0.22 μm) for eliminating any residual impurities and subsequently dried at an elevated temperature (60 °C) resulting in the formation of the TNP precursor.

To prepare a homogeneous solution, boric acid (~1.5 g, J.T. Baker, Radnor, PA, USA, purity: 99.5%) was added to the TNP precursor (~3 g) and the solution was dispersed in dimethylformamide (DMF, ~50 mL, MACRON, Radnor, PA, US, purity: 99.8%) and then the beaker was placed in an ultrasonic bath (Delta company, Taipei, Taiwan, model: DC400H, maximum power: 400 W, bath temperature: 35 °C) for half an hour. For initiating the solvothermal synthesis procedure, the resulting solution was placed into an autoclave (Teflon-lined, 120 mL, Macro Fortunate Co., Ltd., Taipei, Taiwan) for 12 h at 180 °C. To remove any residue and insoluble impurities, the as-prepared solution (including B-N-codoped GQDs) was filtered through vacuum filtration using a mixed cellulose ester microporous filter (MF-Millipore™ Membrane Filter, Darmstadt, Germany, pore size: 0.22 μm) and subsequently dried using a rotary evaporator (Rightek company, New Taipei, Taiwan, model: VP30; power: 80 W) at an elevated temperature (45 °C). A similar solvothermal technique was repeated for synthesizing N-doped GQDs with no boric acid inclusion during the procedure.

### 3.2. Characterization of B-N-Codoped GQDs

To explore the crystalline structure of B-N-codoped GQD samples, X-ray powder diffraction (XRD) was adopted. The XRD setup was equipped with the Shimadzu LabX Shimadzu, Kyoto, Japan (model: XRD-6000) automated X-ray diffractometer. For analyzing the chemical composition of the samples, XPS (X-ray photoelectron spectroscopy, model: ESCA210/Fiscon VG, AZ, US) was used with a non-linear optimization algorithm developed for fitting the B 1s, C 1s, O 1s, and N 1s spectra. The peaks were deconvoluted by using the computer software (XPSPEAK41, 4.1, Hong Kong, China) with an appropriate full width at half maximum. Subsequently, the microstructure of the as-prepared samples was assessed utilizing the transmission electron microscope (TEM, model: F200s/Talos/FEI, Middlesex County, MA, US).

### 3.3. Electrochemical GOR on GQD Electrodes

To prepare the GQD-coated samples, we adopted a commonly used drop-coating technique which was operated in the liquid phase. Herein, the GQD powder (25 mg) was dissolved within a solution containing 6.5 mL of distilled water and 0.9 mL of Nafion 117 solution (Sigma-Aldrich; St. Louis, MO, US, concentration: 5 wt%). The resulting mixture was ultra-sonicated (Delta company, Taipei, Taiwan, model: DC400H, maximum power: 400 W, bath temperature: 35 °C) for 0.5 h to form a uniform paste. A drop-coating method (3 mL GQD slurry) was performed on an indium tin oxide (ITO, Ruilong company, Miaoli, Taiwan, model: AT35, thickness: 1.1 mm, sheet resistance: 30 $\Omega$ sq$^{-1}$, diamond knife manual cutting) conducting glass to prepare the catalytic electrodes. The recipe for the preparation of GQD slurry was based on our in-house preliminary study. Herein, the Nafion served as not only a conducting medium but also a binder, capable of bonding GQDs with the ITO substrate. Therefore, we concerned two points: (i) Strong adhesion between GQDs and ITO substrate and (ii) the appropriate amount of Nafion allows the GQD surface exposure to the glucose-containing electrolyte. The ITO substrates with an exposed area of 3 × 3 cm$^2$ were cleaned through a rigorous sonicating procedure (Delta company, Taipei, Taiwan, model: DC400H, maximum power: 400 W, bath temperature: 35 °C) in ethanol and deionized water prior to the GQD coating. The as-prepared electrodes (ITO/GQD) were subsequently placed in a vacuum oven (Deng Yng company, New Taipei, Taiwan, model: IACF-DOV40, volume: 60 L) maintained at 80 °C. To explore the catalytic activity of the as-prepared samples, a typical three-electrode electrochemical reactor was used with the saturated Ag/AgCl reference electrode (filling solution: 3 M KCl) and Pt foil serving as the counter electrode. One home-made stainless folder was used to closely connect the ITO/GQD as the working electrode. The three-electrode reactor was filled with a concentrated glucose and NaOH solution (1 M each) and the cyclic voltammetry experiments were conducted with a single-channel potentiostat (Autolab, Herisau, Switzerland, model: 302N/PGSTAT).

### 4. Conclusions

A novel solvothermal technique was developed using the combination of C/N/B precursors for synthesizing B-N-coped GQDs as non-metal catalysts towards GOR. Both N-doped GQD and B-N-codoped GQD catalytic electrodes, with an average particle size of ca. 4.0 nm, contained a high oxidation and amidation level, i.e., N/C ratio: 13.3 at.% and O/C ratio: 21.6-22.2 at.%. The B/C ratio was found to be 3.16 at.%, where the B dopants formed through different bonding types (i.e., N-B, C-B, BC$_2$O, and BCO$_2$) inserted into or decorated on the GQDs nanostructure. The CV profiles of both GQD catalyst electrodes contained (i) direct GOR in the positive scan and (ii) reduction of oxidized GQD sites in the reverse potential scan. The comparison of oxidation current revealed that the catalytic activity of B-N-codoped GQD catalyst was 1.21 times higher than that of N-doped GQD. Therefore, the catalytic activity was significantly enhanced due to the synergistic effect of B and N dopants within the GQD catalysts. Analyzing the Tafel plots, the B-N-codoped-GQD catalyst electrode exhibited a higher exchange current along with the reduced Tafel

slope. The B-N-codoped GQD electrodes demonstrate an excellent catalytic activity and substantially improve the kinetics of the glucose oxidation reaction promoting the facile electron transfer. Accordingly, the novel design of GQD catalytic electrodes demonstrated in this work set the stage for synthesizing robust, high-performance, and inexpensive catalysts as a replacement for precious metal nanocatalysts commonly used for a variety of electrochemical devices (e.g., bio-sensors and fuel cells).

**Author Contributions:** Conceptualization, C.-T.H., R.-S.J., and K.D.K.; methodology, S.G.; resources, S.G., C.-T.H., R.-S.J., and K.D.K.; data curation, C.-P.K. and C.-C.F.; software: C.-P.K. and C.-C.F.; validation, C.-T.H. and R.-S.J.; writing—original draft preparation, C.-T.H. and Y.A.G.; writing—review, validation, and editing, C.-T.H., R.-S.J., K.D.K., and Y.A.G.; visualization, C.-P.K., C.-C.F., and Y.A.G.; supervision, C.-T.H., R.-S.J., and K.D.K.; project administration, S.G., C.-T.H., and R.-S.J.; funding acquisition, S.G., C.-T.H., and R.-S.J. All authors have read and agreed to the published version of the manuscript.

**Funding:** The major financial support for this work was provided by the Ministry of Science and Technology of Taiwan (MOST 108-2221-E-155-036-MY3). The authors also acknowledge the partial financial support provided by the Chang Gung Memorial Hospital, Linkou, Taiwan (Chang Gung Medical Foundation, Taiwan, grant number CMRPD2E0082). S.G. also thanks the partial support received from the Program for Innovative Research Team in Science and Technology in Fujian Province University (IRTSTFJ), as well as the Talents Introduction Program of Xiamen University of Technology (YKJ19018R) and the Natural Science Foundation of Fujian Province (no. 2020J01288).

**Institutional Review Board Statement:** The study did not require ethical approval.

**Informed Consent Statement:** "Not applicable" for studies not involving humans.

**Data Availability Statement:** Data is contained within the article.

**Conflicts of Interest:** The authors declare no conflict of interest.

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
