# Peer review of "Electrocatalytic Oxidation of Glucose on Boron and Nitrogen Codoped Graphene Quantum Dot Electrodes in Alkali Media"

_catalysts, doi:10.3390/catal11010101_

Round 1

Reviewer 1 Report

Please, see the attachement. 

Reviewer 2 Report

In general the paper is well methodological written and relevant references to the previous works in the field are well documented. In my opinion, the article may be accept for publication in the present form. 

Reviewer 3 Report

This paper describes electrochemical glucose oxidation catalyzed by B-N codoped graphene quantum dots.  Development of GOR catalyst without transition-metal is significant because direct oxidation glucose fuel cell supplys the necessary energy for implanted medical devices. In this paper, authors have reported B-N co-doped GQD prepared by a simple solvothermal method shows relatively high catalytic activity for electrochemical GOR.  So, this manuscript is basically of potential interest for the readership of Catalysts. However, performance of the B-N co-doped GQD as a GOR catalyst is not clear.  Therefore, I would like to decide my recommendation of this paper after the authors have addressed the following points.

Major points

  1. To clarify catalytic performance of N-doped and B-N co-doped GQD, authors must show the results of controlled-potential electrolysis of a glucose solution at several potentials in main text. Time variation of current density (A cm-2) clearly indicates catalytic activity and stability of the catalysts.
  2. Figure 6 of P8. Authors should explain the reason why the current increase with increase of scan cycle of CV.

Minor points

  1. Figure 6 of P8. To clearly compare N-doped and B-N codoped GQD, current density of Figure 6a and 6b should be displayed by same scales.

Round 2

Reviewer 1 Report

The previous comments/suggestions have been properly amended. 

Small typos need to be corrected on line 176 and 177 (PDF#...)

Reviewer 3 Report

According to reviewer's request, authors fully revised the manuscript.

Therefore, I recommend publication of this manuscript as an article of Catalysts.
